# Plasmids in the human gut reveal neutral dispersal and recombination that is overpowered by inflammatory diseases

Alvah Zorea [1,2,3], David Pellow[4], Liron Levin[5], Shai Pilosof[2,3], Jonathan Friedman [6], Ron Shamir[4] & Itzhak Mizrahi [1,2,3] ✉

Plasmids are pivotal in driving bacterial evolution through horizontal gene transfer. Here, we investigated 3467 human gut microbiome samples across continents and disease states, analyzing 11,086 plasmids. Our analyses reveal that plasmid dispersal is predominantly stochastic, indicating neutral processes as the primary driver of their wide distribution. We find that only 20-25% of plasmid DNA is being selected in various disease states, constraining its distribution across hosts. Selective pressures shape specific plasmid segments with distinct ecological functions, influenced by plasmid mobilization lifestyle, antibiotic usage, and inflammatory gut diseases. Notably, these elements are more commonly shared within groups of individuals with similar health conditions, such as Inflammatory Bowel Disease (IBD), regardless of geographic location across continents. These segments contain essential genes such as iron transport mechanisms- a distinctive gut signature of IBD that impacts the severity of inflammation. Our findings shed light on mechanisms driving plasmid dispersal and selection in the human gut, highlighting their role as carriers of vital gene pools impacting bacterial hosts and ecosystem dynamics.

Plasmids play a vital role in bacterial evolution and gene transfer, but limited studies have explored their natural distribution and function across ecosystems[1–3]. Understanding their ecological roles can provide insights into microbial community dynamics and evolution[4–7]. Plasmids can be categorized into three lifestyles: conjugative, mobilizable, and non-mobilizable[8]. Additionally, phages have been identified as major contributors to the dissemination of non-mobilizable plasmids[9]. Plasmids contain genes required for their own maintenance, such as DNA replication and mobility genes, as well as accessory genes that confer advantages to bacterial hosts under selective pressures, such as antibiotic and heavy metal resistance, organic compound degradation, and virulence[10,11].

Plasmids facilitate the transfer of genetic material, including antibiotic-resistance genes[12–14]. That being said, it is important to broaden our understanding of other functional elements transferred via plasmids. Specifically, investigating the forces driving plasmid segment recombination across ecosystems can provide deeper insight into the ecological functions and implications of these genetic elements within microbial communities. There is limited research on the distribution of plasmids, especially in the human gut, across different geographical regions and disease states[15].

Understanding how ecological and geographical barriers influence plasmid dispersal and recombination is crucial for gaining deeper insights into their implications within microbial communities. In this context, there are two opposing forces for the dispersal of entities

[1]National Institute of Biotechnology in the Negev, Ben-Gurion University of the Negev, 8410501 Be'er Sheva, Israel. [2]Department of Life Sciences, Ben-Gurion University of the Negev, 8410501 Be'er Sheva, Israel. [3]The Goldman Sonnenfeldt School of Sustainability and Climate Change, Ben-Gurion University of the Negev, 8410501 Be'er Sheva, Israel. [4]Blavatnik School of Computer Science, Tel Aviv University, 69978 Tel Aviv, Israel. [5]Bioinformatics Core Facility, llse Katz Institute for Nanoscale Science and Technology, Ben-Gurion University of the Negev, 8410501 Be'er Sheva, Israel. [6]Institute of Environmental Sciences, Hebrew University, Rehovot, Israel. ✉e-mail: imizrahi@bgu.ac.il

across ecosystems: neutral (stochastic) and selective (deterministic). Neutral theory[16] assumes that stochastic dispersal and drift are responsible for community assembly, hence the composition at a local scale is shaped by random dispersal from the global pool. "Nonneutral" niche theory[17] suggests that the abundance of microbial species is influenced by environmental factors, thus communities with similar ecological conditions are likely to have similar microbial compositions. Several studies on microbial environments have found that both selective and neutral forces shape microbial communities in various environments, such as hot springs, wastewater treatment plants, and human microbial environments[18-20]. These forces were used to determine ecosystem functions, such as in a study of lung microbiomes, where the neutral model was used to differentiate between healthy and diseased lungs[18].

To the best of our knowledge, there are no studies that have examined the effect of neutrality on plasmid dispersal. Deeper insight in this regard is fundamental, as it may contribute to our understanding and improve predictions of plasmids' functional contribution to their environment, allowing us to manipulate or selectively target them to modify their functionality to suit our needs.

In this study, we focused on both healthy individuals and those with diseases associated with gut microbiome dysbiosis, including Glucose-metabolism Related Diseases (GRD), Inflammatory Bowel Disease (IBD), and obesity[21-23]. Our research explores patterns of plasmid dispersal among individuals across these disease states (defined as both healthy and diseased individuals) and geographical

locations and investigates the interconnection between plasmids' coding capacity and lifestyle. Using a neutral community model, we explored the diversity and mobility of plasmids in these environments and analyzed the impact of selective environments on the distribution of plasmid gene content. Finally, we constructed a plasmid similarity network to examine the extent and restrictions of plasmid segment dispersal and recombination tendencies between individuals. We found that neutral forces primarily govern plasmid and plasmid segment dispersal, while mobility and inflammatory diseases can influence this process by driving plasmid segment recombination and selecting for disease-related functions, even in the presence of geographical barriers. Our findings reveal the dynamics of plasmid dissemination, including their segments and the carried functions, and the factors driving their spread in gut bacterial communities.

## Results
### Gut inflammation is reflected in the plasmid-to-species richness ratio

We analyzed 3467 human gut samples from 26 datasets across different geographies spanning four different continents (Fig. 1A, Supplementary Table 1 and Supplementary Table 2). A total of 38,383 plasmids were assembled using SCAPP[24], a plasmid assembler that classifies plasmids according to the presence of plasmid genes or the presence of sequences with high similarity to known plasmid sequences. These were verified using other tools for plasmid classification and plasmid gene annotation (PlasForest[25], MOB-suite[26],

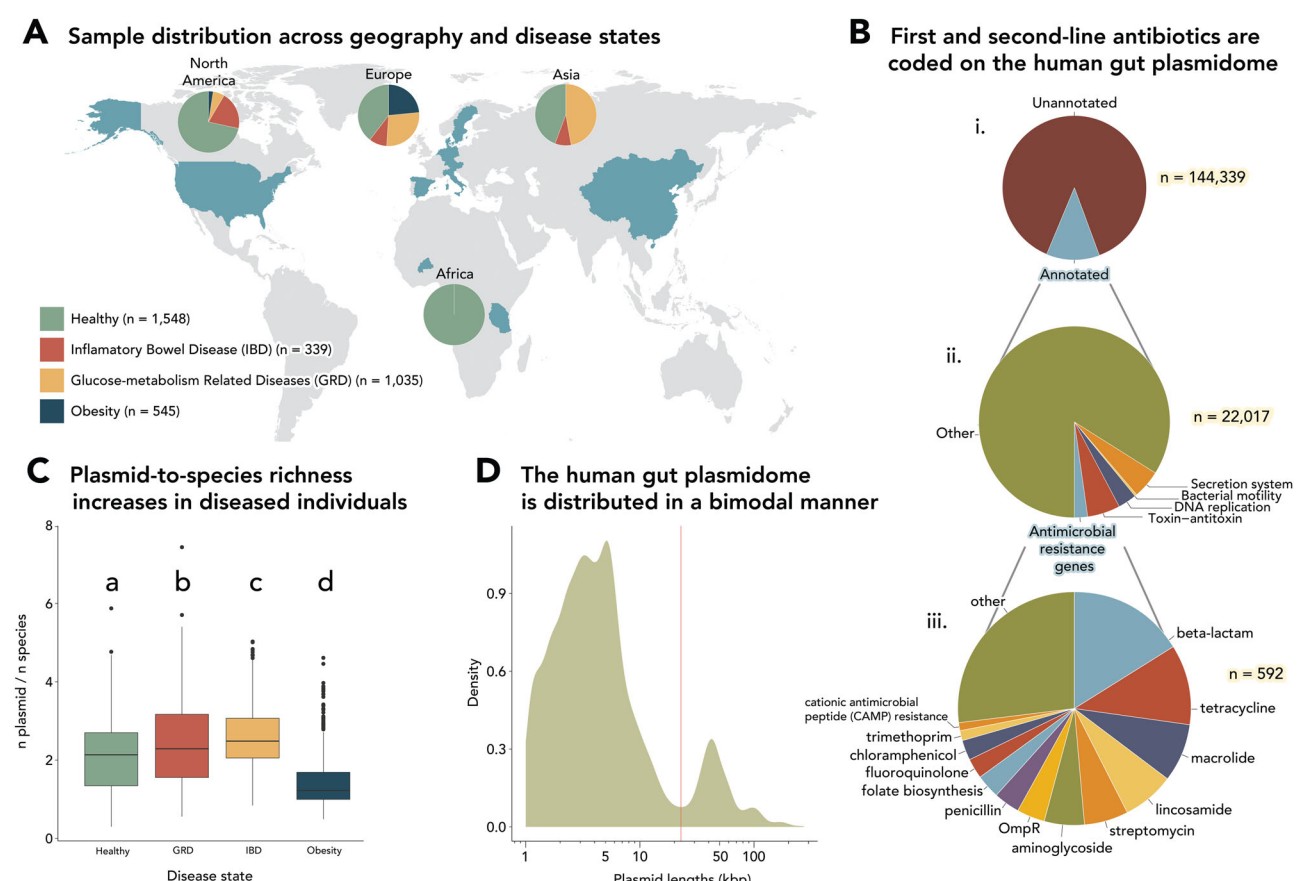

**Fig. 1 | Plasmids assembled in this study and the functions they encode for.**
**A** Proportions of disease states of samples analyzed in this study, per continent.
**B** The proportions of (i) annotated ORFs, (ii) KEGG Orthology pathways and Brite levels, and (iii) antimicrobial resistance genes found on assembled plasmids.
**C** Plasmid/species richness distributions within each disease state (two-sided Wilcoxon rank-sum test, false discovery rate (FDR) corrected $p = 0.00021$).

Midlines of boxplots represent the median, boxes the interquartile range (25th to 75th percentile), and whiskers the range of data. $n = 1548$ healthy, 339 IBD (Inflammatory Bowel Disease), 1035 GRD (Glucose-metabolism Related Diseases), and 545 obese individuals. **D** A linear–log plot depicting the plasmid length distribution in kilobase pairs (kbp). The red line represents the local trough between peaks.

Blastn[27], PlasClass[28] and an in-house plasmid gene database[29], Supplementary Fig. 1A and "Methods"), and were reduced to 11,086 unique plasmids after removing duplicates. Of all open reading frames (ORFs) encoded on these plasmids, 15.25% were annotated, including DNA replication proteins, toxin-antitoxin and secretion systems, plasmid mobility genes, and antimicrobial resistance (AMR) genes (Fig. 1Bi, ii). These genes conferred resistance to a wide range of antibiotics, including some of the most relevant families in clinical practice (i.e., Cephalosporins and Aminoglycosides) and might represent the common use of antibiotic agents as part of the westernized lifestyle (Fig. 1Biii and Supplementary Fig. 1B).

Previous studies have shown that diseases, particularly those involving inflammation of the gut, are associated with reduced gut microbiota richness[30,31]. Our analysis extends this finding to plasmid richness, referring to the variety of plasmids, clustered at 95% identity over 95% of the larger plasmid's length in a given sample, as we found that plasmid richness was significantly higher in healthy samples compared to diseased samples (Wilcoxon rank-sum test, $p < 0.01$, Supplementary Fig. 1C). This is in line with a recent metagenomic study that pointed to a reduced plasmid richness in IBD and *Clostridium difficile* patients[32]. However, when plasmid richness was normalized by microbial richness by calculating the ratio of plasmids to microbial species richness, this phenomenon was reversed, with higher ratios observed in both Inflammatory Bowel Disease (IBD) and Glucose-metabolism Related Diseases (GRD) compared to healthy and obese groups (Wilcoxon rank-sum test, $p < 0.001$, Fig. 1C and Supplementary Fig. 1D). The decrease in plasmid richness could suggest that the inflammatory diseases exerted a type of selection on the plasmidome and microbiome, while the increase in the ratio of the plasmid to species richness might suggest that per species, the versatility of the mobile gene pool increases, highlighting the plasmidome as a reservoir of functions that may relate to the diseased environment. Our analysis also revealed a bimodal distribution of plasmid lengths (Fig. 1D), with a smaller proportion of larger plasmids, consistent with previous research[33]. This could indicate a potential negative selection against medium-sized plasmids and suggest an evolutionarily stable size range.

## Plasmid lifestyle determines the functions they carry

We categorized plasmid lifestyle based on the presence or lack of mobility genes, into mobilizable (1027) and non-mobilizable (10,059) plasmids. There was a significantly higher frequency of mobilizable plasmids in individuals with IBD and GRD compared to healthy or obese individuals (IBD 2.63%, GRD 2.04%, healthy 1.69%, and obesity 1.64%, pairwise chi-square test, $p < 0.001$, 95% CI, Fig. 2A). These findings could suggest that mobilizable plasmids may have a more significant role in horizontal gene transfer (HGT) in diseased environments associated with inflammation, a notion supported by previous research that has documented an acceleration of HGT in inflamed states[34]. A KEGG pathway enrichment analysis of plasmids revealed distinct patterns regarding the functional potential and gene content of mobilizable vs. non-mobilizable plasmids (hypergeometric test, $p < 0.0001$, gene ratio > 0.1, Fig. 2B). Non-mobilizable plasmids were enriched with maintenance-related functions, including homologous recombination. In addition, non-mobilizable plasmids were enriched with accessory functions that confer advantages to the bacterial host within the gut ecosystem, such as carbon utilization enzymes, energy harvesting enzymes, and enzymes involved in cofactor and vitamin production, all belonging to the "Metabolic pathways" pathway (Fig. 2B). Among these accessory genes, we detected important enzymes that reflect the gut ecosystem's central functions, including those involved in sugar metabolism and fermentative energy harvesting pathways, such as cellobiose phosphorylase [EC:2.4.1.20], rhamnulokinase [EC:2.7.1.4], xylulokinase [EC:2.7.1.17], acyl-CoA dehydrogenase [EC:1.3.8.7], and butyryl-CoA dehydrogenase

[EC:1.3.8.1]. Non-mobilizable plasmids also carried genes involved in cofactor and vitamin production, such as nicotinate-nucleotide adenylyltransferase [EC:2.7.7.18], an enzyme that promotes oxidative phosphorylation and improves host energy utilization efficiency. Although non-mobilizable plasmids were enriched with more pathways essential for their maintenance, the average size of mobilizable plasmids was significantly larger than that of non-mobilizable plasmids even after excluding mobilization genes from mobilizable plasmids (11,098.42 vs. 10,276.74, respectively, Wilcoxon rank-sum test, $p < 0.001$, Fig. 2C). This could indicate that on average, mobilizable plasmids may carry additional genes, which could be reflected in different functional potentials. Indeed, our analysis of the distribution of AMR genes on mobilizable versus non-mobilizable plasmids showed a significant difference, with mobilizable plasmids carrying more AMR genes compared to non-mobilizable plasmids (Kolmogorov–Smirnov test, $D = 0.05$, $p < 0.05$, Fig. 2D). Specifically, mobilizable plasmids were found to be six times more likely to carry AMR genes compared to non-mobilizable plasmids (odds ratio = 6.14). These findings suggest that the mobilization lifestyle of plasmids influences their gene content and functional potential, where mobilizable plasmids are more likely to carry systems important for their own transfer, as well as accessory AMR genes that are generic and thus could be used by various microbial hosts regardless of their ecological niche. In contrast, non-mobilizable plasmids carry accessory genes that are ecologically relevant and directly linked to their microbial host's ecological niche. It is important to acknowledge that our efforts to understand the selective forces acting on plasmids are challenged by the limited annotation of genes, and that selection driven by adaptive traits encoded by non-annotated genes is likely an important factor for the detected plasmid distribution as well.

## Plasmid dispersal is neutral and is affected by disease state and plasmid mobilization lifestyle

To unravel the forces driving plasmid dispersal, we employed a neutral community model (NCM)[16]. This model assumes that plasmids are dispersed across individuals in proportion to their abundance in the regional pool. The $R^2$ value of the NCM provides an estimate of the fit of plasmid dispersal to the neutral model, with higher values implying greater random dispersal, and lower values suggesting selective factors influence plasmid distribution.

Plasmids' dispersal fits the neutral model with a relatively high $R^2$ value in our dataset, suggesting predominantly neutral dispersal ($R^2 = 0.5$, Fig. 3A). Here, we applied higher stringency in our analysis, where we added two more filtration steps. Across the samples, a depth cutoff was established at 1% of the lowest plasmid abundance observed in the sample with the lowest read depth, and a minimum read coverage threshold of 70% per plasmid was applied, resulting in 3966 plasmids.

We next investigated the impact of plasmid lifestyle on dispersal (Fig. 3B). Mobilizable plasmids ($n = 123$) exhibit dispersal rates more consistent with neutrality ($R^2 = 0.82$), compared to non-mobilizable plasmids ($n = 3843$) with $R^2 = 0.52$. This observation was corroborated by randomly selecting 123 non-mobilizable plasmids in 1000 iterations and comparing their $R^2$ values to that of the mobilizable plasmids ($p = 0.03$, Supplementary Fig. 2A). This suggests that non-mobilizable plasmids experience stronger selection pressure, likely due to their selection together with their host cells and limited dispersal capacity, resulting in reduced neutral dispersal. In contrast, a mobilizable lifestyle allows plasmids greater autonomy, potentially relieving selective pressure and enabling more neutral dispersal. Our analysis showed a good fit of plasmid dispersal to the neutral model in all disease states (GRD $R^2 = 0.49$, IBD $R^2 = 0.41$, obesity $R^2 = 0.59$, healthy $R^2 = 0.49$, Supplementary Fig. 2B), indicating that plasmids may maintain a neutral dispersal, even in diseased environments. Nevertheless, the noticeable decrease in $R^2$ values of plasmid dispersal in IBD patients

**A** Mobilizable plasmids are more prevalent in individuals with IBD and GRD

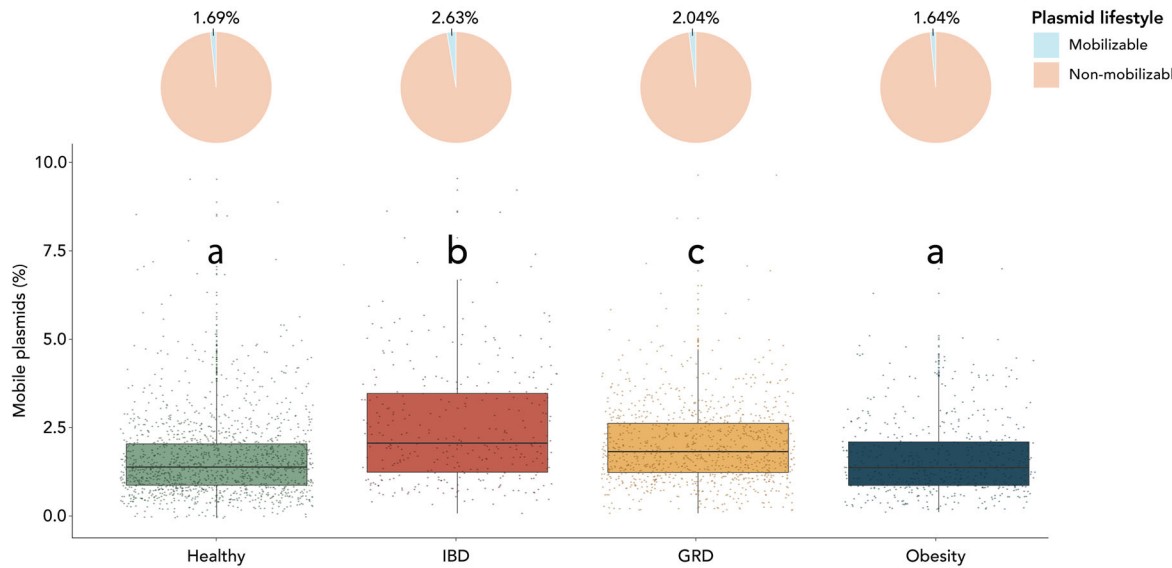

**B** Plasmids' mobility lifestyle defines the functional cargo they carry

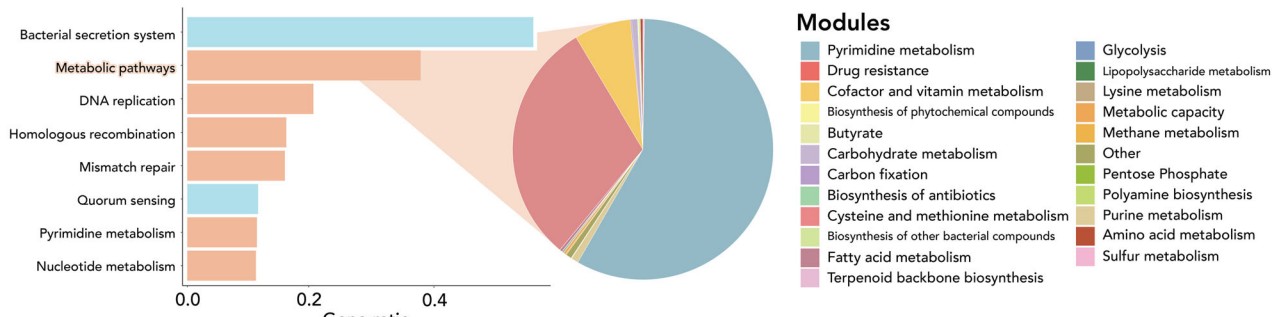

**C** Plasmids' mobility lifestyle is connected to their length

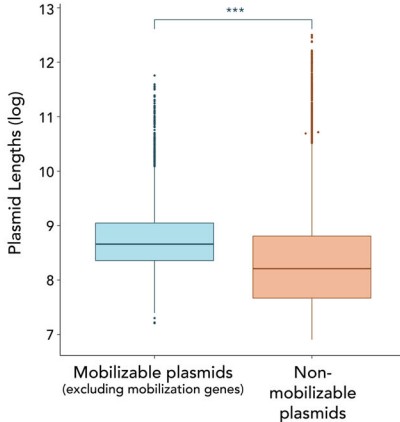

**D** AMR genes are more prevalent on mobilizable plasmids

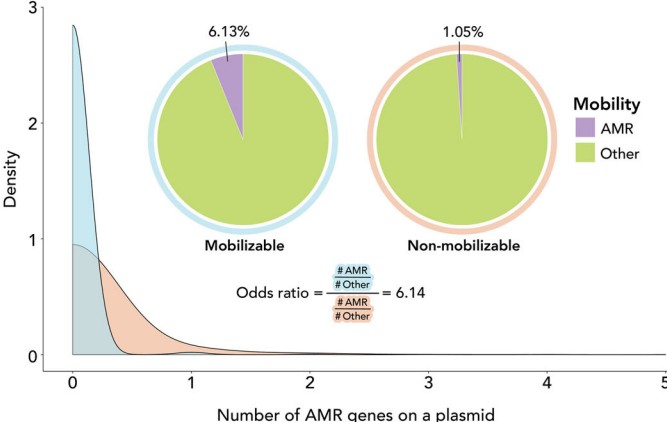

**Fig. 2 | Plasmid lifestyle dictates their distribution and functionality.**
**A** Proportions and distributions of mobilizable and non-mobilizable plasmids in individuals as a function of their disease state (two-sided Wilcoxon rank-sum test, false discovery rate (FDR) corrected $p = 5.9e{-}6$). Midlines of boxplots represent the median, boxes the interquartile range (25th to 75th percentile), and whiskers the range of data. $n = 1548$ healthy, 339 IBD (Inflammatory Bowel Disease), 1035 GRD (Glucose-metabolism Related Diseases), and 545 obese individuals. **B** Pathways enriched on mobilizable and non-mobilizable plasmids and the proportions of modules found within the KEGG "Metabolic pathways" category (hypergeometric test, FDR corrected $p < 0.0001$, gene ratio $>0.1$). Colors represent the different plasmid lifestyles, while gene ratios represent the prevalence of the pathways within each plasmid lifestyle. **C** Mobilizable vs. non-mobilizable plasmid lengths (log scale), (two-sided Wilcoxon rank-sum test, $p = 2e{-}16$). Midlines of boxplots represent the median, boxes the interquartile range (25th to 75th percentile), and whiskers the range of data. $n = 1027$ mobilizable plasmids and 10,059 non-mobilizable plasmids. **D** Distributions and ratios of AMR (antimicrobial resistance) gene count on plasmids as a function of plasmid mobilization lifestyle.

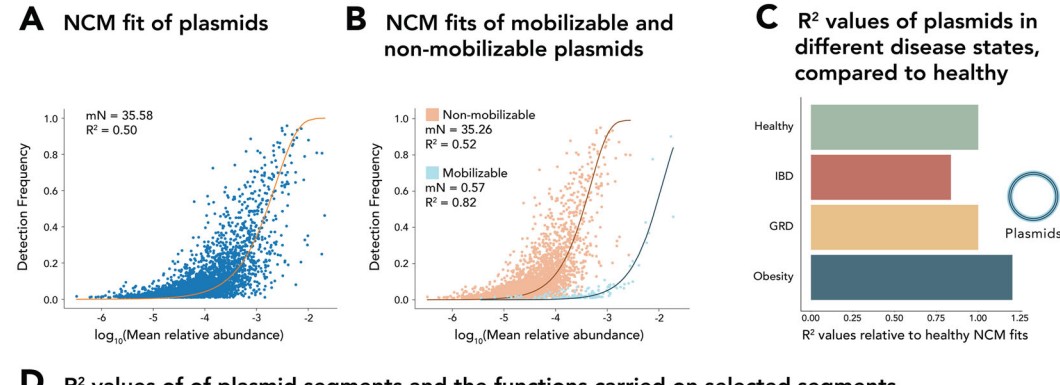

**A** NCM fit of plasmids

**B** NCM fits of mobilizable and non-mobilizable plasmids

**C** $R^2$ values of plasmids in different disease states, compared to healthy

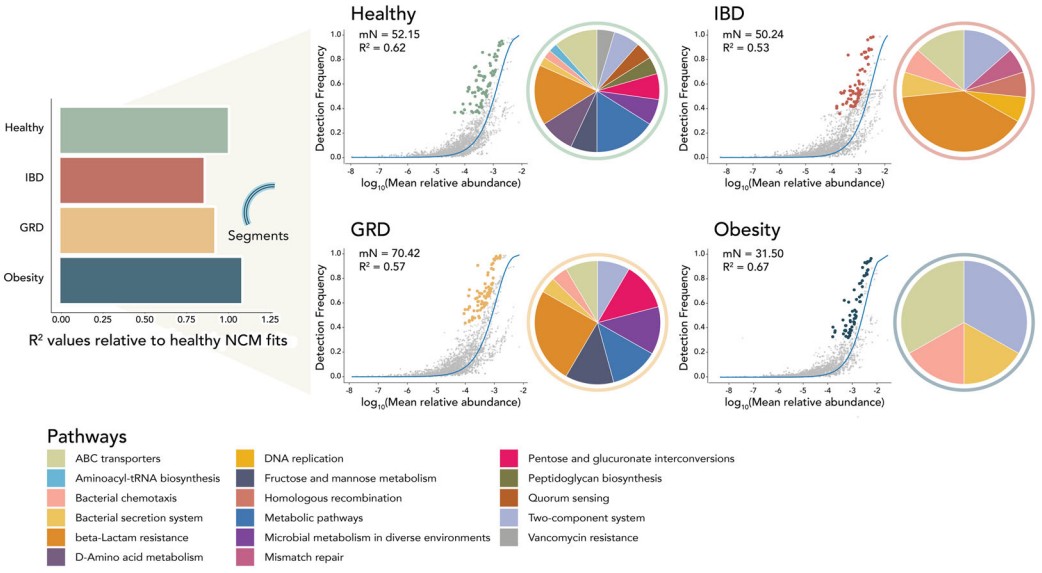

**D** $R^2$ values of of plasmid segments and the functions carried on selected segments

Pathways
- ABC transporters
- Aminoacyl-tRNA biosynthesis
- Bacterial chemotaxis
- Bacterial secretion system
- beta-Lactam resistance
- D-Amino acid metabolism
- DNA replication
- Fructose and mannose metabolism
- Homologous recombination
- Metabolic pathways
- Microbial metabolism in diverse environments
- Mismatch repair
- Pentose and glucuronate interconversions
- Peptidoglycan biosynthesis
- Quorum sensing
- Two-component system
- Vancomycin resistance

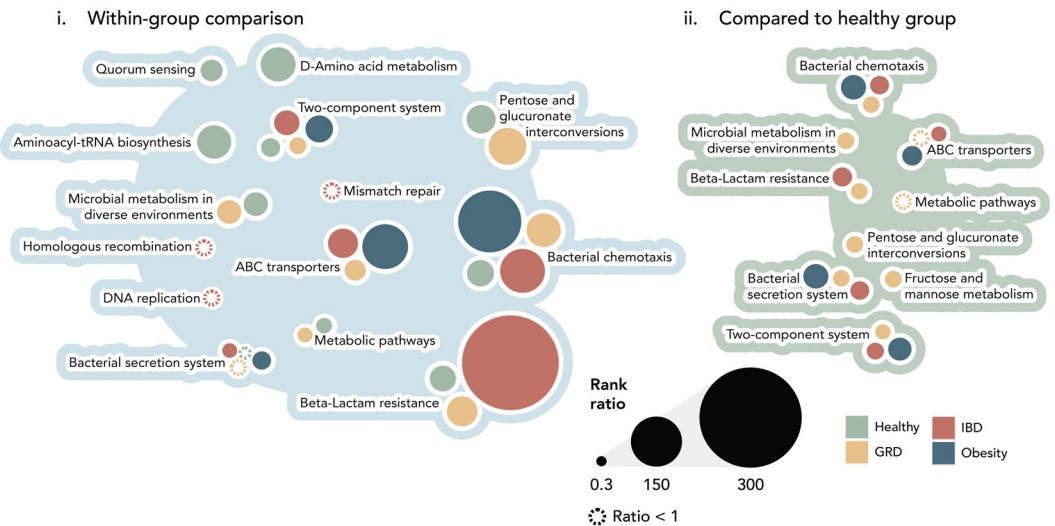

**E** Ratios of functions on selected segments

i. Within-group comparison

ii. Compared to healthy group

Rank ratio

0.3    150    300

Ratio < 1

Healthy    IBD
GRD    Obesity

compared to healthy individuals indicates that plasmid distribution is more neutral in healthy samples than in IBD (a 16% decrease for IBD compared to healthy individuals, Fig. 3C).

We divided our dataset into mobilizable and non-mobilizable plasmids obtained from healthy individuals, as well as from those with obesity, IBD, and GRD, and compared their $R^2$ values of NCM fits (Supplementary Fig. 2C). This hints that plasmid lifestyle was the primary factor influencing neutrality, with mobilizable plasmids showing more neutral dispersal than non-mobilizable plasmids. Common to both lifestyles, was the decrease in neutrality for plasmids originating from individuals with IBD compared to healthy individuals, highlighting this disease as a more selective one. Specifically, mobilizable plasmids in both IBD and GRD environments were dispersed in a less neutral manner than in healthy individuals (IBD $R^2 = 0.75$, GRD

**Fig. 3 | Neutral community model fits of plasmids and segments as a function of disease state and plasmid lifestyle.** Fit of the neutral community model (NCM) of all plasmids (**A**) and of the mobilizable (blue) and non-mobilizable (orange) plasmids, separately (**B**). Each dot represents a plasmid, and the solid lines indicate the best fit to the NCM. $R^2$ values measure the goodness of fit to the neutral model and the mN values are the migration rates from global to local patches (metacommunity size times immigration). Comparison of plasmid (**C**) and plasmid segment (**D**) dispersal in healthy and diseased environments, represented by ratios of $R^2$ values ($R^2$/healthy $R^2$). **D** NCM fits of plasmid segments, separated by disease state. Each dot represents a segment. Selected segments that occur more frequently than predicted by the model among individuals are shown in different colors, corresponding to the disease state (>95% prevalent and >95% confidence interval around the neutral fit). The pie chart adjacent to each plot shows the distribution of the KEGG pathways present on these selected segments. IBD Inflammatory Bowel Disease, GRD Glucose-metabolism Related Diseases. **E** Comparison of the odds ratios of KEGG pathways carried on selected segments in each disease state as compared to all segments within the same disease state (i) e.g., the odds ratio within IBD individuals of the beta-lactam resistance on selected plasmids is 284 times higher when compared to its ratio in the non-selected plasmid population in IBD individuals, and compared to selected segments in healthy individuals (ii) e.g., the beta-lactam resistance odds ratio on selected plasmids in IBD is 3.52 times higher when compared to selected plasmids in healthy individuals. The colors of the bubbles represent the disease state of the selected segments that are being compared, while their sizes represent their odds ratio value. Bubbles with dashed borders indicate odds ratios with values less than 1.

$R^2 = 0.77$, healthy $R^2 = 0.82$) whereas non-mobilizable plasmids were dispersed in a less neutral manner in IBD when compared to healthy (IBD $R^2 = 0.46$, healthy $R^2 = 0.49$).

Given that plasmid recombination drives plasmid adaptation[33,35], we examined the dispersal and the evolutionary forces acting on plasmids by studying plasmid segments, defined as stretches of plasmid DNA of at least 1 kbp in length and at least 80% identity between two plasmids. Similar to what we observed in plasmids, we saw a tendency for less neutral dispersal among individuals with IBD and GRD compared to healthy individuals, with a 15% decrease in $R^2$ values for IBD samples and an 8% decrease for GRD samples compared to healthy individuals, IBD $R^2 = 0.53$, GRD $R^2 = 0.57$, healthy $R^2 = 0.62$, Fig. 3D). We investigated the functional composition of specific plasmid segments that exhibited deviations from the neutral model fit and were widely present among individuals, suggesting that these segments were undergoing selection (Fig. 3D and Supplementary Fig. 3). The functions carried on these segments significantly differed from the overall distribution within each disease state (Fisher's exact test, IBD GRD and healthy $p < 0.001$, obesity $p < 0.05$, Fig. 3Ei). Notably, beta-lactam resistance was the predominant function among selected plasmid segments across conditions, except in obesity. In individuals with IBD, it accounted for 40% of selected functions, in GRD it was 25%, and in healthy samples, it constituted 16%. The odds ratios for beta-lactam resistance genes on selected segments compared to the overall distribution within each disease state were 284 for IBD, 15 for GRD, and 10 for healthy individuals. Moreover, the odds ratios of beta-lactam resistance genes on the selected segments were higher in IBD and GRD compared to healthy individuals, with a ratio of 1.76 for GRD vs. healthy and 3.52 for IBD vs. healthy (Fig. 3Eii). These findings indicate that antibiotic usage strongly drives plasmid segment selection in the human gut, particularly in individuals with IBD and GRD.

Segments with selection-driven dispersal across all diseases exhibited a higher prevalence of the "ABC transporters" pathway compared to other non-selected segments within the same disease, except for healthy individuals (odds ratio: IBD 15.03, GRD 4.77, obesity 11.68). Additionally, the observed odds ratios were higher when compared to selected segments within healthy individuals in the case of IBD and obesity, but not GRD (odds ratio: IBD 1.2, obesity 3.9). The "ABC transporters" pathway in our dataset included genes from the lantibiotics permease system, commonly carried on plasmids and play important roles in maintaining gut homeostasis[36]. This suggests that segments encoding for these AMR systems confer a specific advantage to their microbial hosts[37].

## Disease state selects for ecologically relevant functions via plasmid segment recombination

To delve deeper into the relationship between plasmid segment dispersal, disease state, and geography, we generated a network in which samples (nodes) were connected if they shared plasmid segments (edges, Fig. 4Ai). Using this approach, we could analyze and visualize the intricate relationships between humans via segment recombination and dispersal across the plasmid population. The network revealed distinct patterns in segment sharing among individuals with IBD and GRD, indicating a predominant random recombination pattern, together with selective forces that are at play within these diseases (Fig. 4Aii). Specifically, individuals with IBD and GRD shared a higher number of plasmid segments within their groups compared to between the groups (Wilcoxon rank-sum test, $p < 0.0001$, Fig. 4Bi, ii, and Supplementary Fig. 4A). The ratio of plasmid segment sharing within disease states compared to between disease groups was 1.36 for IBD and 1.19 for GRD, while for healthy individuals, the ratio was lower, ~1 (Fig. 4Bii). This observation suggests that specific segments are selected for and shared across plasmids within the IBD and GRD groups.

We further investigated the cross-continental edges within each disease state and found that individuals with IBD and GRD shared more plasmid segments per person within their respective groups than healthy individuals, suggesting that the selective pressures in these diseases overpower geographic ones (mean of IBD 174.15, GRD 238.95, and healthy 158.42, Wilcoxon rank-sum test, $p < 0.01$, Supplementary Fig. 4B). To identify non-random plasmid segment sharing within each disease state, we generated a null model through 10,000 simulations by randomly permuting edges while preserving node degrees and comparing it to the observed network to determine significance (with samples belonging to the obesity disease group disregarded due to their limited geographic origin). Strong patterns of sharing plasmid segments with ecologically relevant functions were observed among individuals with IBD and GRD ($p < 0.0001$, FDR corrected). This sharing appeared to overcome geographical barriers, as significant non-random cross-continental segment sharing was exclusively detected in these two diseases and not among healthy individuals (Fig. 4Aii). In IBD, 56.1% of the cross-continental edges showed non-random sharing, while in GRD, only 3.77% exhibited non-random sharing. Healthy individuals did not exhibit any significant non-random cross-continental connectivity, and other non-random connections were observed within continents, highlighting the impact of geography on plasmid segment dispersal.

Moreover, our findings suggest that the observed patterns are primarily influenced by large and mobile plasmids, supported by odds ratios exceeding 1 compared to the overall dataset. This aligns with our detection of more mobilizable plasmids within the disease states of IBD and GRD (Fig. 2A). In individuals with IBD and GRD, significant segments shared across continents were primarily associated with mobile plasmids (odds ratio of 7.14 and 3.96, respectively, Supplementary Fig. 4C) and large plasmids (odds ratio of 1.98 and 1.27, respectively).

Similar significant connectivity within disease states was seen when creating networks based on segments of 500 bp in length (corresponding to the predominant length of open reading frames found on plasmids) and permuting them 10,000 times (Supplementary Fig. 4D, Ei, $p < 0.0001$, FDR corrected). Specifically, individuals with IBD, GRD, and healthy controls exhibited significant connectivity

**A** Segment-sharing network

  i.   Schematic illustration of the plasmid segment network

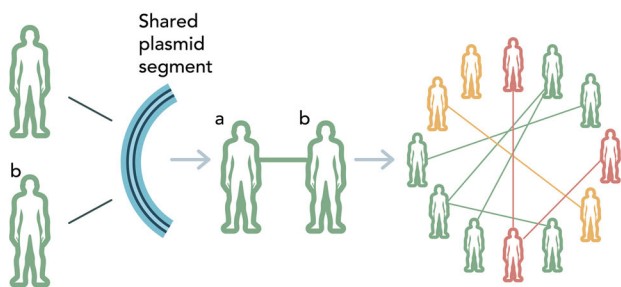

  ii.  Intercontinental segment sharing in IBD and GRD and not in healthy individuals

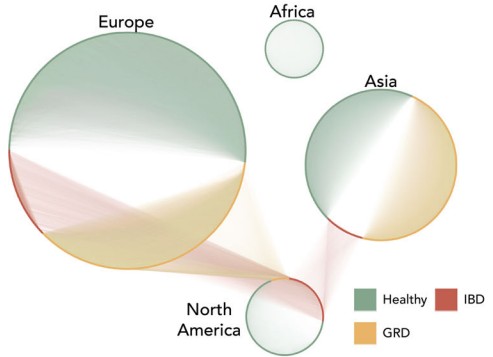

**B** Segment sharing in different disease states

  i.   Schematic illustration of segment sharing within and between disease states

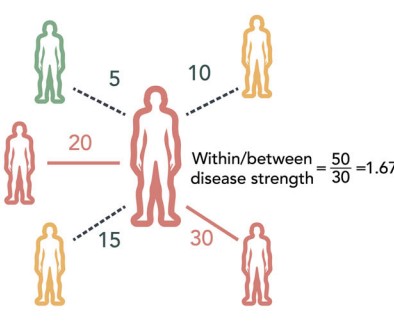

  ii.  Disease-specific Segment sharing patterns in IBD and GRD

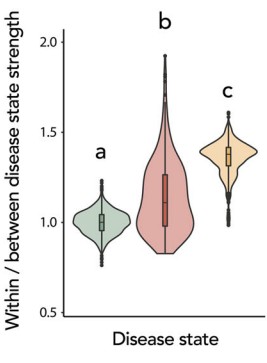

  iii. Disease-associated functional potential

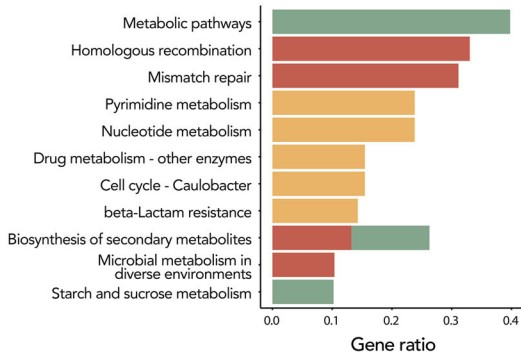

**C** Intercontinental segment sharing

  i.   Schematic illustration of intercontinental segment sharing

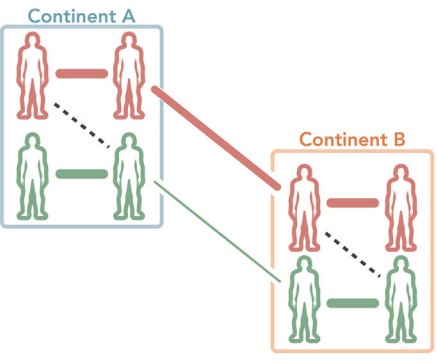

  ii.  Selected functions are shared across continents in IBD and GRD

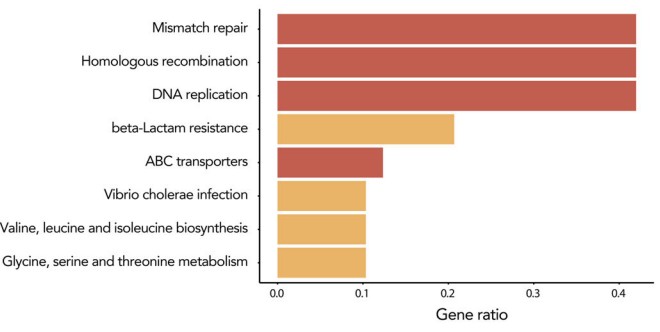

**Fig. 4 | Selection acts on segments shared within disease groups and overcomes spatial barriers. A** A schematic (i) and actual (ii) representation of the similarity network, where nodes represent humans, and the edges connecting individuals of the same disease state symbolize shared plasmid segments. The weights of these edges represent the number of shared segments between any two individuals. For visualization purposes only, edges that appear in this network connect between humans if they share at least 100 segments (weight > 100) and are part of a group of disease states and continents that exhibit significant connectivity compared to a null model. Nodes are grouped by their continent of origin and are colored according to their disease states. IBD Inflammatory Bowel Disease, GRD Glucose-metabolism Related Diseases. **B** (i) A diagram illustrating the definition of an individual's connectivity (determined by their strength, i.e., the sum of edge weights) with others of the same disease state (within-disease strength, represented by solid lines) compared to those of different disease states (between-disease strength, represented by dashed lines). Numbers adjacent to the edges represent their weights (i.e., the number of shared plasmid segments between two individuals). (ii) A violin plot showing the ratio of strengths between individuals within the same disease state and those from different disease states (two-sided Wilcoxon rank-sum test, false discovery rate (FDR) corrected $p = 2e{-}16$). Boxplots indicate the median and quartiles, with whiskers reaching up to 1.5 times the interquartile range. The violin plot outlines illustrate kernel probability density, i.e., the width of the shaded area represents the proportion of the data located there. $n = 1548$ healthy, 339 IBD, 1035 GRD, and 545 obese individuals. (iii) Functions that are enriched on significantly connected segments within each disease group, as determined by comparing their frequencies to the null model (hypergeometric test, FDR corrected $p < 0.01$, gene ratio > 0.1). Colors represent the different disease states, while gene ratios represent the prevalence of the pathways within each disease state. **C** (i) A schematic diagram depicting increased cross-continental connectivity between diseases compared to healthy. (ii) Functions that are enriched on significantly connected plasmids within IBD and GRD individuals across continents (hypergeometric test, FDR corrected $p < 0.01$, gene ratio > 0.1). Colors represent the different disease states, while gene ratios represent the prevalence of the pathways within each disease state.

within continents. Additionally, individuals with IBD and GRD demonstrated significant cross-continental connectivity, with GRD significantly connected between Europe and North America, and IBD showing significant connections between North America and Europe, as well as North America and Asia. In contrast, healthy individuals did not show any significant cross-continental connectivity. Similar patterns were observed for networks based on segments of 1000 bp with 90 and 95% similarity ($p < 0.005$, FDR corrected, Supplementary Fig. 4Eii, iii).

By comparing the original and permuted networks and analyzing the segment frequencies across individuals of each disease state, we identified segments that displayed significant connectivity within each respective disease state. Within each disease group, 20–25% of the overall plasmid segments were significantly shared (20% in IBD, 23% in obesity and 25% in both GRD and healthy individuals). Analysis of the functions carried on these selected segments revealed three enriched pathways in healthy individuals, four in IBD, and five in GRD (hypergeometric test, $p < 0.01$, gene ratio > 0.1, Fig. 4Biii). In individuals with IBD, the pathways "Mismatch repair" and "Homologous recombination" were significantly enriched. Additionally, in these individuals, we observed the presence of the enzyme aerobactin synthase in two significantly enriched pathways: "Biosynthesis of secondary metabolites" and "Microbial metabolism in diverse environments". Aerobactin synthase is an iron-related virulence factor known to induce inflammation in patients with Crohn's Disease. It has been found to be overexpressed in virulent *E. coli* strains that are more prevalent in IBD patients compared to commensal strains and is considered the main gut microbiome signature in IBD[38]. This enzyme, typically plasmid-encoded, is an iron-scavenging siderophore and serves as a fitness determinant, providing a competitive advantage to pathogenic bacteria over non-carriers of this system[39]. In GRD, the "beta-Lactam resistance" pathway was enriched, which is directly linked to diabetes development[40]. These findings provide further evidence that the highly connected plasmid segments within disease environments may carry ecologically relevant functions for their microbial host and are linked to the disease.

This observed effect was due to recombining segments impartial from the microbial host (Supplementary Fig. 5), suggesting that segments, rather than entire plasmids, are likely the basic entities under selection. These findings suggest that the movement of microbes or whole plasmids is not the primary driver of plasmid functional spread, but rather recombination events occurring among plasmid entities. Moreover, this strongly indicates that the selective pressures imposed by the diseases IBD and GRD drive the sharing and recombination of plasmid segments within these groups, even across great distances.

Analyses of specific functions enriched on cross-continental plasmid segments revealed that many of these segments carry advantageous functions linked to the pathology of the disease (hypergeometric test, $p < 0.01$, gene ratio > 0.1, Fig. 4Ci, ii). Among individuals with IBD, we identified 830 segments that were significantly shared across continents, with 525 carrying annotated functions. Notably, the "ABC transporters" pathway was enriched in cross-continental segments in IBD samples, including proteins of the SitABCD system involved in manganese/iron transport systems, previously identified on plasmids[41]. Iron deficiency anemia is a common complication in IBD, where iron leaking from the intestinal environment is scavenged mainly by pathogenic bacteria in the gut, transforming commensal gut microbes into pathobionts and inducing inflammation. Accordingly, iron was found by many studies to be the main factor in determining the inflammation state in IBD by favoring pathogenic enterobacteria in the gut[42–44].

We identified 358 segments significantly shared across continents among individuals with GRD, with 187 carrying annotated functions. These segments were enriched with KEGG pathways such as "beta-Lactam resistance", consistent with our previous finding of higher

proportions of this pathway on selected segments in GRD (Figs. 3E and 4Cii). Moreover, the "Vibrio cholerae infection" pathway, including the Zona occludens toxin (Zot), was enriched on cross-continental segments in GRD patients. This toxin is the main determinant of peptide transfer through the intestinal epithelium[45] and accordingly, affects intestinal permeability to insulin by modifying tight junctions that restrict the transfer of these peptides[46]. These results support the involvement of specific selective forces in plasmid segment dispersal and sharing, prioritizing essential functions for microbial hosts to cope in these environments, such as iron scavenging, and can directly relate to the disease pathology. Importantly, evidence of segment selection was also observed within healthy individuals (Fig. 4Biii), indicating that these selective pressures may not be exclusive to diseased environments.

## Discussion

Plasmid studies have traditionally focused on antimicrobial resistance gene dispersal under antibiotic-selective conditions[1,12–14]. However, the broader role of plasmids and their selective environments has been overlooked, and in vitro studies may not fully capture their natural dynamics. Consequently, while plasmids are often perceived as under constant selection, in natural conditions, selection may not be constant or prevalent, suggesting that plasmids are exposed to stochastic forces and disperse neutrally.

In this study, we sought to determine whether plasmid dispersal aligns with the predictions of the neutral theory of biodiversity[16] or deviates from them, indicating nonrandom patterns driven by selective pressures. Our findings challenge prevailing beliefs, showing that plasmid dispersal is predominantly driven by neutral forces, suggesting random dispersal. Additionally, the lifestyle of plasmids affects their dispersal, with mobilizable plasmids showing more neutral tendencies than non-mobilizable ones, suggesting that mobilizable plasmids are better equipped to disperse and colonize in non-selective conditions. It should be noted that further exploration of different subdivisions of plasmid physiology such as plasmid incompatibility groups within each plasmid lifestyle might yield different insights into plasmid dispersal patterns with relation to these subgroups.

Our analysis revealed a bimodal distribution of plasmid lengths, indicating a potential negative selection against medium-sized plasmids and suggesting evolutionarily stable size ranges. This finding may be connected to the functional content, gene burden, and mobilization lifestyle of plasmids. Indeed, we find that the mobilization lifestyle of plasmids significantly impacts their gene content and functional potential. Mobilizable plasmids carry systems that facilitate their transfer between hosts, as well as AMR genes, while non-mobilizable plasmids carry ecologically relevant accessory genes that benefit bacterial hosts within the gut. AMR genes have broader applicability, protecting various hosts regardless of their metabolism, unlike specific metabolic functions that might not fit certain microbial hosts. This pattern can be explained by metacommunity theory paradigms, such as the patch-dynamic paradigm, which proposes a trade-off between dispersal and local dominance in patchy habitats[47]. Accordingly, mobilization systems in mobilizable plasmids increase their dispersal frequency, increasing their chances of survival along with the generic AMR genes they carry. In contrast, non-mobilizable plasmids have limited dispersal, but carry genes that reflect the environment's desirable traits, enabling local dominance due to the advantage they confer to microbial hosts. However, the small sample size of mobilizable plasmids in our study may limit the detection of additional enriched accessory functions.

In individuals with Inflammatory Bowel Disease (IBD) and Glucose-metabolism Related Diseases (GRD), we observed a decrease in neutral dispersal for plasmids and their recombination fragments, indicating higher selection forces within these environments. This aligns with decreased plasmid and microbial species richness in these individuals,

implying higher ecological selection rates in these environments. Interestingly, we observed an increase in plasmid richness per microbial species in these environments, which may suggest an increase in the plasmid reservoir and their functions within phylogenies, mostly true for the non-mobilizable plasmids which constitute the majority of our dataset.

We found that ecologically selected plasmid segments, defined as those that were more prevalent than their relative abundance in the global pool, have a different distribution of functions compared to the overall function pool within their corresponding disease state. This suggested that specific functions were under selection, despite the general neutral dispersal of plasmid segments. We observed an enrichment of beta-lactam resistance in IBD, GRD, and healthy hosts, indicating the impact of antibiotic overuse in human populations[48]. These selective patterns, as well as others, were prominent in GRD and IBD, but less evident or absent in obesity, implying minimal impact of obesity's low-grade inflammation on microbiome composition[49,50].

Network analysis of plasmid recombinating segments across continents and disease states further supported the role of selection acting on specific functions. Increased plasmid segment sharing within IBD and GRD, compared to healthy individuals, indicated selection acting on these segments via HGT[51]. Notably, non-random plasmid segment sharing between continents was observed in individuals with IBD and GRD only, and not healthy individuals. These segments encoded for disease-relevant genes that could provide an advantage to the bacterial host, further supporting the notion of selection on plasmid segments carrying specific functions in diseased environments. On these shared segments in individuals with IBD, we identified genes associated with the scavenging and transport of iron, which has been previously linked to the pathogenesis of IBD by exacerbating dysbiosis[52]. Collectively, our results suggest that segment dispersal via selection or random process is mainly driven by recombination events and not whole plasmid mobilization or movement together with the microbial host itself. This is seen in the higher turnover within and between continents of the segments when compared to the plasmids and microbes as well as the fact that the edges in our network are composed of segments and not whole plasmids.

Our study provides strong evidence that plasmid dispersal in human gut systems is predominantly random, with many plasmids dispersing neutrally. However, under harsher selection pressures, specific plasmid segments with ecological functions are selected, reflecting the environment's most important traits. These segments act as reservoirs of fundamental functions in each ecosystem, potentially supporting the overall health and stability of the microbiome.

Furthermore, our findings have implications for understanding the spread and evolution of antibiotic resistance, challenging the deterministic view of plasmid selection in antibiotic resistance spread and emphasizing the unpredictable nature of resistance gene dissemination. This highlights the need to consider the stochastic aspect of plasmid dispersal in combating antibiotic resistance. The study underscores the importance of studying plasmids in diverse environments to gain insights into their ecology and evolution[53].

This study may also contribute to resolving the plasmid paradox, where plasmids persist in bacterial populations despite the inherent fitness costs they impose[54,55]. It suggests that plasmids persist not because they confer a fitness advantage to their hosts, but due to the neutral dispersal and stochastic forces on both the plasmid and its host. Overall, our findings represent a paradigm shift in our understanding of plasmids and their role in the spread and evolution of accessory functions.

## Methods
### Datasets
Metagenomic paired-reads of 3588 samples were downloaded from the National Center for Biotechnology Information's (NCBI) Sequence Read Archive (SRA) from a total of 26 Bioprojects (Supplementary Table 1). These data spanned different continents and diseases associated with dysbiosis (Fig. 1A and Supplementary Table 2). Samples with read depths below 2 million were discarded from the analyses, resulting in 3467 samples with read depths between 2 and 86 million reads.

### Plasmid assembly
Paired-end reads were trimmed and cleaned using Trim Galore v2.6[56] and assembled into contigs by Megahit v1.0.3[57]. Sequentially, a total of 38,383 plasmids were assembled by SCAPP v0.1.4[24]. To deduplicate the plasmids, we employed a clustering approach that kept the larger of two plasmids if their identity (as determined by BLASTn v2.10.1+[27]) was above 95% and covered at least 95% of the larger plasmid, which reduced the data to 11,086 non-redundant plasmids. We conducted additional analyses on these 11,086 predicted plasmids and plotted these results using the "UpSetR"[58] R package (Supplementary Fig. 1A). We used additional tools, including MOB-suite[26], which annotates plasmid genes such as rep, mob, oriT, and mpf, Blastn[27], which we used to annotate additional plasmid genes within the nr database, PlasForest[25], a random forest classifier to identify contigs of plasmid origin, PlasClass[28], a k-mer based sequence classifier which uses a set of standard classifiers trained on the most current set of known plasmid sequences for different sequence lengths achieving higher F1 scores in classifying sequences from a wide range of datasets, and finally, an in-house plasmid gene database (available on GitHub[29]). Altogether, all identified plasmids are either predicted as such or carry plasmid genes according to at least one of the employed methods above.

A lot of reads are lost in this process of read assembly into contigs, contig assembly into plasmids, and further filtration steps done by SCAPP to reduce false positive plasmids. To compensate, reads were mapped to the non-redundant plasmids using BBmap v38.86[59], and their abundance per sample was determined using Metabat2 v2.12.1[60]. The read coverage of plasmids in each sample was computed by SAMtools mpileup v1.10[61]. Plasmids were considered present in a sample if they had at least 70% coverage (i.e., reads mapped at least once over 70% of the plasmid length), reducing our data to 10,605 plasmids. To compensate for uneven read depths between samples, a depth cutoff was determined as 1% of the lowest plasmid abundance in the sample with the lowest read depth (~2 million reads), refining our data to 3966 plasmids.

In all subsequent analyses, plasmid sequences were duplicated to avoid cases where genes were split into two segments in the represented output sequence, due to random linearization during the de-novo assembly process. The output was then corrected for this duplication by removing repetitive results.

### Plasmid and metagenome annotation
We utilized MOB-suite v3.0.3[26] to classify plasmids into 10,059 non-mobilizable and 1027 mobilizable (105 conjugative and 922 mobilizable). We regarded both "mobilizable" and "conjugative" plasmids as "mobilizable" plasmids, as the only way to determine whether a "hitchhiker" mobilizable plasmid can transfer horizontally is by examining the presence of a conjugative plasmid within the same cell, on the single-cell level. Following the plasmid filtration process described earlier, this was reduced to 123 and 3843 mobilizable and non-mobilizable plasmids, respectively. Enrichment analyses were performed by comparing the functions associated with each plasmid lifestyle to the overall distribution of functions on plasmids (hypergeometric test, false discovery rate (FDR) corrected $p < 0.0001$, gene ratio > 0.1)

To assess that the observed difference in plasmid lengths between the mobilizable and non-mobilizable groups wasn't due to the presence of the relatively large backbone mobility genes[62], we subtracted

these gene lengths from the length of mobilizable plasmids when comparing plasmid lengths between the two groups.

A total of 144,339 Open Reading Frames (ORFs) were predicted by Prokka v1.12[63]. Annotations were achieved using anvi'o v7.1[64] by converting plasmids and segments to an anvi'o database (using anvi-gen-contigs-database) and annotating them with the Kyoto Encyclopedia of Genes and Genomes (KEGG) KOfam database version 4[65] (using anvi-run-kegg-kofams), resulting in a total of 22,016 KOs on plasmids (15.25% of the predicted and clustered ORFs) and 12,567 on segments. The annotated entities were then used as input to the program anvi-estimate-metabolism, with the parameter --output-modes set to both hits and modules to get the enzyme annotations and their corresponding metabolic modules. These steps were all run in parallel using the NeatSeq-Flow workflow platform[66]. Subsequently, we identified 592 KOs belonging to the "Antimicrobial resistance genes" Brite level. Of these, 252 AMR genes were validated with the Resistance Gene Identifier (RGI) v6.0.1, based on the Comprehensive Antibiotic Resistance Database (CARD) v3.2.5[67] (Supplementary Fig. 1B). These were cross-referenced with Brite annotations, resulting in AMR genes. Read taxonomies were determined by MetaPhlAn v4.0.3[68].

### Plasmid segmentation
Segments were defined as a stretch of at least 1000 bp of DNA repeating in at least two samples with at least 80% identity, as determined by a reciprocal BLASTn of plasmids against their duplicated selves, resulting 8234 segments. These were then clustered by 80% identity with coverage of at least 90% length of both segments using cd-hit-est v4.8.1[69,70], and filtered by plasmid coverage, refining the data to a total of 6138 segments, across 1754 plasmids (44.23% of all plasmids, average length: 3033 bp). Notably, 3.4% of these shared segments also included full plasmid sequences that recombined into larger plasmids. These rare cases may in fact be a result of gene gain or loss and we cannot differentiate between the two.

### Neutral community model
The relative abundances of plasmids were fit to a neutral community model (NCM)[16]. This model was modified to account for the large population sizes of prokaryotic communities and allows for the incorporation of competitive advantage[71,72]. We examined to which degree the abundances of plasmids in different diseases fit this near-neutral model. Nm indicates metacommunity size times immigration. When comparing mobilizable and non-mobilizable neutral models, we corrected for the uneven sample size by randomly subsampling the non-mobilizable plasmids to match the sample size of the mobilizable plasmids. This was done 1000 times for comparison. Segments under selection within each disease state were defined by being in the 5% highest frequency and 5% highest deviation values from the neutral fit (determined by their distance from the fit on the $y$-axis, Supplementary Fig. 3).

### Network construction and analyses
Using the "igraph"[73] R package, we constructed the network of interactions across the human cohort by connecting two humans if they share at least one plasmid segment. In the resulting network based on these segments, we observed a total of 1,931,343 interactions between humans (out of a possible 4,501,500, network density: 43%). To identify significantly connected subgroups, we conducted a 10,000-fold randomization process by permuting the network using the "BiRewire"[74] R package. This package is specifically designed for bipartite networks, ensuring that the degree of each node in the original network is conserved while maximizing randomization. Subsequently, we reconstructed the segment-sharing network based on these permutations. We then analyzed the non-random connections

within each disease state and compared connectivity across and within continents to assess the effect of geographic barriers ($p < 0.0001$, FDR corrected). To eliminate any bias, we excluded samples from the obesity disease group, which mainly came from one continent only. The network was visualized using Cytoscape[75]. For visualization purposes only, we reduced the network complexity by filtering out edges with weights less than 100 (meaning humans that shared less than 100 segments were not connected by an edge in the figure) and filtered out edges connecting humans of different disease states. All analyses and statistics were done on the original unfiltered network. Significant edges within each disease state were determined by comparing their frequency in the original network with that in all permuted networks ($p < 0.05$, FDR corrected). Enrichment analyses were performed by comparing the functions associated with these significant segments to the overall distribution of functions observed on segments (hypergeometric test, FDR corrected $p < 0.01$, gene ratio >0.1, Fig. 4Biii). In addition, significant cross-continental disease groups (specifically IBD and GRD) were analyzed for significantly connected edges within them. These significant edges were defined by comparing their frequency between significantly connected continents in the original network with that in all permuted networks ($p < 0.05$, FDR corrected). Enrichment analyses were performed by comparing the functions associated with these significant segments to the overall distribution of functions observed on segments within the respective diseases (hypergeometric test, $p < 0.01$, gene ratio >0.1, Fig. 4Cii).

### R packages
Statistical analyses were carried out using R 3.5.1[76]. Data manipulation was achieved using "tidyverse"[77] and "dplyr"[78] R packages. Graphs were generated using the R package "ggplot2"[79], including the map displayed in Fig. 1A, statistics were plotted with "ggpubr"[80] and the graphics were modified using "ggh4x"[81] and "ggtext"[82]. All enrichment tests of KEGG Orthology pathways were achieved using a hypergeometric test and a false discovery rate correction for multiple testing, with "clusterProfiler"[83]. Jaccard distances were calculated using the "vegan"[84] R package. Chi-square, Fisher's exact, and Wilcoxon rank-sum tests, as well as linear models and correlations, were calculated using the "stats" R package. All p-values of multiple comparison analyses were corrected accordingly (FDR).

### Statistics and reproducibility
With the exception of samples with low read depths, no data were excluded from the analyses. All statistical analyses conducted in this paper are detailed in the previous sections.

### Reporting summary
Further information on research design is available in the Nature Portfolio Reporting Summary linked to this article.

## Data availability
All the utilized metagenomes in this study are publicly accessible on NCBI: PRJEB17784, PRJNA339012, PRJNA356102, PRJEB18755, PRJNA196801, PRJNA290729, PRJNA690543, PRJEB12947, PRJEB7774, PRJEB7949, PRJEB10878, PRJNA328899, PRJNA321058, PRJEB15371, PRJNA305507, PRJEB2054, PRJEB1786, PRJEB12124, PRJNA319574, PRJNA422434, PRJNA278393, PRJEB4336, PRJEB1220, PRJNA324129, PRJNA299502, PRJNA361402. Details of accession numbers, along with paper references, are outlined in Supplementary Table 1. The relevant metadata, plasmid sequence files, and in-house plasmid gene database are all available on GitHub[29].

## Code availability
The scripts to execute the main analyses conducted in this study are available on GitHub[29].

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

## Acknowledgements

This work was funded by grants from the German-Israeli Project Cooperation (DIP 2476/2-1), the European Research Council (ERC 866530), and the Israel Science Foundation (ISF 1947/19) to I.M. Funding was also provided by the Israel Science Foundation (grant No. 1281/20) and the Human Frontiers Science Program (award RGY0064/2022) to S.P. A fellowship from the Edmond J. Safra Center for Bioinformatics at Tel-Aviv University supported D.P. The Israel Science Foundation (grant No. 2206/22) and Len Blavatnik and the Blavatnik Family Foundation supported R.S. The authors thank Daphne Perlman for the scientific illustrations and the assistance with graphic design in all figures of this paper. Figure 4a–c features a silhouette created by our graphic designer, Daphne Perman, using Adobe Illustrator.

## Author contributions

A.Z. conducted the analyses, consolidated the data, and wrote the paper. D.P. contributed to the analyses and provided writing assistance. L.L. contributed to the analyses. S.P. offered guidance on network analyses. J.F. provided advice on neutral fit analyses and wrote the corresponding scripts. R.S. provided writing assistance and secured funding. I.M. participated in the analyses, consolidated the data, wrote the paper, conceptualized the ideas, and secured the funding.

## Competing interests

The authors declare no competing interests.
