## [Peer Review File · Nature Communications]

REVIEWER COMMENTS

Reviewer #1 (Remarks to the Author):

In their study Zorea et al. aim to answer the question of whether plasmid dispersal and recombination is neutral in healthy and disease cohorts using neutral community models. Their data and model are compelling and this is an interesting way to approach the question of dispersal.

My main concern is with their starting data. The methods that the researchers used to identify plasmids and whether the biases in the methods use underscore their results. For example, the researchers used SCAPP, which is largely based on identifying circular DNA from de Bruijn graphs and was mainly benchmarked against two other assembly tools. Especially for the small plasmids, these are likely incomplete. We find that additional quality control is necessary to annotate and categorize plasmids. Among the ~10k plasmids deemed non-mobilizable, are the researchers sure that these are in fact plasmids? Especially the smaller ones that are between 1-5kb? Do they have partition machinery or relaxases? Assessing the completeness and fidelity of their plasmids is important for their downstream conclusions.

On the flip side, promiscuous or highly recombinogenic plasmids may have assemblies that are unnaturally diverse or contain more genes due to misassembly (combinations of segments that may not actually appear in a single plasmid together but retain similar pieces). This may mar conclusions such as that in figures 2C and D.

The authors perform an analysis of plasmid fragments instead of full plasmids. This is useful for correcting the above problem (albeit large plasmids will be over-counted compared to smaller ones). The 1 kb cutoff seems perhaps arbitrary. Have the authors tried to cluster similar regions on plasmids to identify more appropriate units?

Given the above comments, do the results between disease/healthy cohorts hold when considering only large or only mobile plasmids?

Minor comments:

Please define what plasmid richness means in the text.

Is there bias introduced in the depth cutoff for the plasmid dispersal model (line 196)? Read coverage of 70% of a very large sample is a lot more reads than a depth coverage for a small plasmid. This would be further biased by potential misassemblies (composite assemblies) for large plasmids. How many plasmids were filtered based on these criteria, in which categories?

Reviewer #2 (Remarks to the Author):

This research investigates the central role of plasmids in bacterial evolution through horizontal gene transfer. Plasmids are identified metagenomic data using excising bioinformatic pipeline on > 3000 human gut microbiome samples. The study unveils that stochastic processes predominantly drive plasmid dispersal but approx. 1/4 of plasmids are selectively distributed in disease states.

Specific plasmid segments are shaped by selective pressures tied to mobility, antibiotics, and inflammatory gut diseases. Remarkably, these elements are shared within groups of individuals with similar health conditions, emphasizing their critical role in gut signatures impacting inflammation.

The study sheds light on plasmid dynamics in the human gut, highlighting their importance as carriers of vital gene pools potentially influencing bacterial hosts and ecosystems. The research's novelty lies in its exploration of plasmid distribution and function across diverse ecosystems, offering new insights into potential drivers for microbial evolution and community dynamics.

The manuscript is well written and easy to follow for a broad audience.

In summary, I have listed a few comment below for minor adjustments of the manuscript and recommend acceptance of this manuscript for publication in Nature communications.

This study significantly advances our understanding of plasmid dissemination dynamics and their impact on gut bacterial communities. Especially the concept of neutral forces driving plasmid dispersal is a novel finding that challenges previous assumptions, promising better predictions, and manipulation of plasmid functionality.

General comment:

1) Since the study is entirely based on retrospective correlation analyses on metagenomic data, the proposed selective drivers causing the observed patterns are purely observational. Several statements in the results section should therefore be phrased as hypothesis rather actual findings.

2) As seen in other plasmidome studies the large majority (85%) of ORF encoded on the plasmids could not be annotated, emphasizing the novelty of the identified plasmid pool. However, it is also a major problem for the attempts in this study to explain the distribution of plasmids based on selection for the few successfully annotated genes. Selection driven by adaptive traits encoded by non-annotated genes is likely an important factor for the detected plasmid distribution. This should be stated more clearly in the section on "Plasmid lifestyle determines the functions they carry". The interpretation of the results in this section should be seen in the perspective that only 15 % of the ORF's are included in these analyses and the remaining 85 % may not support these findings.

3) The author observed a higher R² value for the fit of the Neutral Community Model (NCM) for non-mobilizable plasmids compared to mobilizable plasmids (Fig 3B). The graph seems to indicate that the group of non-mobilizable plasmids (which consist of most of the plasmids), is a heterogeneous group that may be further sub-divided into various plasmid types e.g., replicon types. Each plasmid type could then undergo a separate NCM fitting where some non-mobilizable plasmids might have higher R² value. It should therefore be stated that different pattern might be seen if different plasmid subdivisions had been used.

4) The methods for defining plasmid segments rely on two clustering definitions:

i) Clustering plasmids based on identity > 95% and coverage \geq 95%.

ii) Defining plasmid segments as stretches of plasmid DNA at least 1kbp in length with a minimum of 80% identity between two plasmids.

The rationale for these breakpoints is not clear to me. I believe utilizing more stringent criteria in step 2 than in step 1 for clustering the plasmids would enhance the identification of plasmid segments likely resulting from recombination rather than descent from common ancestors.

We would like to thank the reviewers for their comments to modify and improve the manuscript and for their invaluable insights. We genuinely appreciate the time and effort dedicated to the review process, and we are pleased to present the revisions made in response to the constructive feedback.

These revisions include expanding our verification processes of assembled plasmids in our dataset by cross-referencing them to other existing plasmid tools. Additionally, in response to the suggestion to explore different criteria for plasmid segmentation from both reviewers, we conducted extensive analyses, including examining plasmid ORF lengths, which highlighted the prevalence of 500 bp segments. In response, we conducted additional network analyses and permutations based on these segments, including repeating the network creation and 10,000 permutations. We also repeated this process using other segment criteria (1000 bp with 90% identity and 1000 bp with 95% identity). These analyses showed similar trends to the original data, underscoring their robustness, and have been added to the revised manuscript. We took additional steps to provide clarity and be more careful in the wording of our conclusions.

In summary, the collective insights from both reviewers have significantly enriched and strengthened the manuscript. We believe that the revisions made in accordance with your suggestions contribute substantially to the clarity, validity, and impact of our study.

REVIEWER COMMENTS

Reviewer #1 (Remarks to the Author):

In their study Zorea et al. aim to answer the question of whether plasmid dispersal and recombination is neutral in healthy and disease cohorts using neutral community models. Their data and model are compelling and this is an interesting way to approach the question of dispersal.

My main concern is with their starting data. The methods that the researchers used to identify plasmids and whether the biases in the methods use underscore their results. For example, the researchers used SCAPP, which is largely based on identifying circular DNA from de Bruijn graphs and was mainly benchmarked against two other assembly tools. Especially for the small plasmids, these are likely incomplete. We find that additional quality control is necessary to annotate and categorize plasmids. Among the ~10k plasmids deemed non-mobilizable, are the researchers sure that these are in fact plasmids? Especially the smaller ones that are between 1-5kb? Do they have partition machinery or relaxases? Assessing the completeness and fidelity of their plasmids is important for their downstream conclusions.

On the flip side, promiscuous or highly recombinogenic plasmids may have assemblies that are unnaturally diverse or contain more genes due to misassembly (combinations of segments that may not actually appear in a single plasmid together but retain similar pieces). This may mar conclusions such as that in figures 2C and D.

We thank Reviewer #1 for your thoughtful evaluation of our study and acknowledge the importance of addressing biases in metagenomic plasmid assembly. In response to your

constructive feedback, we have undertaken a rigorous analysis, employing five different tools on our identified plasmid database of 11,086 predicted plasmids. The results of these analyses, stemming from your suggestions, corroborated the existence of the identified plasmids and provided additional depth to our manuscript, strengthening the overall quality of our work.

In the revised manuscript, we used five plasmid tools, including "MOB-suite"¹, which annotates the plasmid genes rep, mob, oriT, and mpf, "Blastn"², which we used to annotate additional plasmid genes within the nr database, "PlasForest"³, a random forest classifier to identify contigs of plasmid origin, "PlasClass", a k-mer based sequence classifier which uses a set of standard classifiers trained on the most current set of known plasmid sequences for different sequence lengths achieving higher F1 scores in classifying sequences from a wide range of datasets⁴, and an in-house plasmid gene database that we now provide as a supplementary material⁵, which we used to annotate plasmid genes. All of the identified plasmids are predicted as such and/or carry plasmid genes according to at least one of the employed methods above (see supplementary figure S1A). Altogether, there are 6,686 plasmids of 1-5 kb in length, which include 2,843 plasmids (43%) with rep genes and or partition machinery, as well as other known plasmid genes such as mob genes. The above analyses were now added to the main text, the Methods, as well as in supplementary figure S1A.

The authors perform an analysis of plasmid fragments instead of full plasmids. This is useful for correcting the above problem (albeit large plasmids will be over-counted compared to smaller ones). The 1 kb cutoff seems perhaps arbitrary. Have the authors tried to cluster similar regions on plasmids to identify more appropriate units?

We appreciate this comment regarding the analysis of plasmid segments. In response to the reviewer's suggestion, we conducted a thorough examination of the size distribution of open reading frames (ORFs) within our plasmid dataset to compare these to the chosen segment sizes. This detailed analysis revealed that a substantial number of plasmid ORFs in our dataset are predominantly under 1000 base pairs, with the most prevalent size being approximately 500 base pairs (See figure Supplementary S4D).

Accordingly, we repeated our network analyses with segments of 500bp. The outcomes of these analyses consistently mirrored the patterns observed with the original data, further corroborating and strengthening our findings. Specifically, we clustered plasmid segments of 500 base pairs and 80% identity, conducting the same main analyses as outlined in our paper. This comprehensive approach involved constructing a network and performing 10,000 permutations to identify significantly connected groups of healthy and diseased cohorts within and between continents, which revealed consistent results in terms of significant connections within different disease categories. Similar to our results in the original manuscript based on segments of 1000bp, individuals with Inflammatory Bowel Disease (IBD), glucose-related diseases (GRD), and healthy controls exhibited significant connectivity within continents. Moreover, individuals with IBD and GRD demonstrated significant cross-continental connectivity, with GRD significantly connected between Europe and North America, and IBD showing significant connections between North America and Europe, as well as North America and Asia, as opposed to healthy individuals that did not show any significant cross-continental

connectivity. The results of these analyses now appear in the main text and supplementary figure S4Ei.

As these new results were essentially identical to our results in the original manuscript, we made a deliberate choice to define shared segments of 1000 base pairs. This decision aligns with a prior study- Shapiro et al. 2023⁶, which defined shared plasmid segments using 1000bp as well. We believe that this clarification addresses the concern raised by Reviewer #1 and strengthens the rationale behind our choice of segment length for the analyses.

Given the above comments, do the results between disease/healthy cohorts hold when considering only large or only mobile plasmids?

Thank you for your insightful comment, which has added another valuable dimension to our findings. In response to your query, we conducted a detailed examination of significant segments connecting individuals with IBD or GRD across continents, specifically assessing their distribution between mobile and non-mobile, as well as large and small plasmids.

Accordingly, we performed odds ratio analyses to ascertain the significance of these distributions, comparing them to the broader distribution of mobile and non-mobile, as well as large and small plasmids in the entire dataset. Our findings suggest that the observed patterns are primarily influenced by large and mobile plasmids, supported by odds ratios exceeding 1 when comparing these distributions to the overall dataset.

In IBD individuals, significant segments connecting across continents were predominantly associated with mobile plasmids (odds ratio of 7.14) and long plasmids (odds ratio of 1.98). Similarly, in GRD individuals, segments significantly connecting across continents were associated with mobile plasmids (odds ratio of 3.96) and large plasmids (odds ratio of 1.27). We have now incorporated these results into the main text and supplementary figure S4C.

Minor comments:

Please define what plasmid richness means in the text.

We added the relevant clarification to the text:

“Our analysis extends this finding to plasmid richness, referring to the variety of plasmids, clustered at 95% identity over 95% of the longer plasmid’s length in a given sample,...” (lines 112-113)

Is there bias introduced in the depth cutoff for the plasmid dispersal model (line 196)? Read coverage of 70% of a very large sample is a lot more reads than a depth coverage for a small plasmid. This would be further biased by potential misassemblies (composite assemblies) for large plasmids. How many plasmids were filtered based on these criteria, in which categories?

We appreciate the reviewer's attention to the coverage cutoff in our plasmid dispersal model, designed to capture representative coverage for plasmids of varying sizes. Following the reviewer's remark, we observed that the percent ratio of long plasmids (>10kb) to short ones in the original unfiltered dataset (22%, 2023 long vs. 9063 short plasmids) is similar and not

significantly different (chi-square test, p value > 0.5) to that of the filtered plasmids based on this criterion (27%, 103 long vs. 378 short plasmids). We have incorporated this information into the Methods section of the manuscript.

Reviewer #2 (Remarks to the Author):

This research investigates the central role of plasmids in bacterial evolution through horizontal gene transfer. Plasmids are identified metagenomic data using excising bioinformatic pipeline on > 3000 human gut microbiome samples. The study unveils that stochastic processes predominantly drive plasmid dispersal but approx. 1/4 of plasmids are selectively distributed in disease states.

Specific plasmid segments are shaped by selective pressures tied to mobility, antibiotics, and inflammatory gut diseases. Remarkably, these elements are shared within groups of individuals with similar health conditions, emphasizing their critical role in gut signatures impacting inflammation.

The study sheds light on plasmid dynamics in the human gut, highlighting their importance as carriers of vital gene pools potentially influencing bacterial hosts and ecosystems. The research's novelty lies in its exploration of plasmid distribution and function across diverse ecosystems, offering new insights into potential drivers for microbial evolution and community dynamics.

The manuscript is well-written and easy to follow for a broad audience.

In summary, I have listed a few comment below for minor adjustments of the manuscript and recommend acceptance of this manuscript for publication in Nature communications.

This study significantly advances our understanding of plasmid dissemination dynamics and their impact on gut bacterial communities. Especially the concept of neutral forces driving plasmid dispersal is a novel finding that challenges previous assumptions, promising better predictions, and manipulation of plasmid functionality.

General comment:

1) Since the study is entirely based on retrospective correlation analyses on metagenomic data, the proposed selective drivers causing the observed patterns are purely observational. Several statements in the results section should therefore be phrased as hypothesis rather actual findings.

We thank Reviewer #2 for the insightful comments and recognize the importance of clarifying the observational nature of our study. Understanding the retrospective correlation analyses conducted on metagenomic data, we fully appreciate the necessity to contextualize specific statements in the Results section appropriately. Accordingly, we have made adjustments to the text to ensure precision and clarity.

2) As seen in other plasmidome studies the large majority (85%) of ORF encoded on the plasmids could not be annotated, emphasizing the novelty of the identified plasmid pool. However, it is also a major problem for the attempts in this study to explain the distribution of plasmids based on selection for the few successfully annotated genes. Selection driven by adaptive traits encoded by non-annotated genes is likely an important factor for the detected plasmid distribution. This should be stated more clearly in the section on "Plasmid lifestyle determines the functions they carry". The interpretation of the results in this section should be seen in the perspective that on only 15 % of the ORF's are included in these analyses and the remaining 85 % may not support these findings.

We agree that the interpretation of our results in the section 'Plasmid lifestyle determines the functions they carry' is based on the ~15% annotated genes and we recognize the potential influence of the ~85% non-annotated genes in driving plasmid distribution. This limitation is now explicitly addressed in the revised text to offer a more balanced perspective:

“It is important to acknowledge that our efforts to understand the selective forces acting on plasmids are challenged by the limited annotation of genes, and that selection driven by adaptive traits encoded by non-annotated genes is likely an important factor for the detected plasmid distribution.” (lines 178-182)

3) The author observed a higher R2 value for the fit of the Neutral Community Model (NCM) for non-mobilizable plasmids compared to mobilizable plasmids (Fig 3B). The graph seems to indicate that the group of non-mobilizable plasmids (which consist of most of the plasmids), is a heterogeneous group that may be further sub-divided into various plasmid types e.g., replicon types. Each plasmid type could then undergo a separate NCM fitting where some non-mobilizable plasmids might have higher R2 value. It should therefore be stated that different pattern might be seen if different plasmid subdivisions had been used.

We appreciate the insightful observation of the reviewer regarding the potential heterogeneity within the different lifestyle groups of plasmids. In light of this, we have included the following statement in the manuscript:

“It should be noted that further exploration of different subdivisions of plasmid physiology such as plasmid incompatibility groups within each plasmid lifestyle might yield different insights into plasmid dispersal patterns with relation to these subgroups.” (lines 427-429)

4) The methods for defining plasmid segments rely on two clustering definitions:

- i) Clustering plasmids based on identity > 95% and coverage \geq 95%.
- ii) Defining plasmid segments as stretches of plasmid DNA at least 1kbp in length with a minimum of 80% identity between two plasmids.

The rationale for this breakpoints is not clear to me. I believe utilizing more stringent criteria in step 2 than in step 1 for clustering the plasmids would enhance the identification of plasmid segments likely resulting from recombination rather than descent from common ancestors.

Thank you for your comment. In response to your recommendation, we included additional analyses to strengthen the robustness of our findings, exploring more stringent criteria for plasmid segmentation, specifically 90% and 95% identity. Following these analyses, the results and conclusions in both criteria remained the same. While the patterns of significance in plasmid sharing varied between the healthy and diseased groups GRD and IBD, there was consistently higher significant segment sharing in diseased vs. healthy groups.

Specifically, at 90% identity, GRD displayed similar segment-sharing patterns within and between continents as observed with 80% similarity in the original manuscript. In contrast, IBD showed no cross-continental segment sharing but significant within-continent sharing across all three continents. That being said, healthy groups significantly shared segments within Europe and Asia but not North America, indicating higher sharing within the disease groups.

At 95% identity, similar trends were observed, except that GRD exhibited significant cross-continental segment sharing across Europe and Asia as well, adding further significance to the intercontinental segment sharing within diseases.

We retained the 80% identity criterion for plasmid segmentation, aligning with a plasmid segment sharing analysis done in a prior study⁶. We added the results for the 90% and 95% thresholds to the main text and supplementary figure (Fig S4Eii and S4Eiii).

Additionally, in response to your comment, we conducted a meticulous analysis of plasmid ORF sizes, revealing a notable prevalence of ORFs of 500bp in length (Figure S4D). Subsequently, we replicated our entire analysis using this segment size. The results and conclusions exhibited remarkable robustness, remaining essentially unchanged. After thorough consideration, we made the informed decision to uphold the 1000bp plasmid size for our study. This choice is anchored in the robustness of our findings at the 500bp size, ensuring methodological continuity. Importantly, the analysis and conclusions drawn from this exploration are now integrated into the main text and supplementary figure S4Ei. We trust that these clarifications and additional analyses positively contribute to the overall strength and clarity of our work.

1. Robertson, J. & Nash, J. H. E. MOB-suite: software tools for clustering, reconstruction and typing of plasmids from draft assemblies. *Microb Genom* **4**, (2018).
2. Camacho, C. *et al.* BLAST+: architecture and applications. *BMC Bioinformatics* **10**, 421 (2009).
3. Pradier, L., Tissot, T., Fiston-Lavier, A.-S. & Bedhomme, S. PlasForest: a homology-based random forest classifier for plasmid detection in genomic datasets. *BMC Bioinformatics* **22**, 349 (2021).
4. Pellow, D., Mizrahi, I. & Shamir, R. PlasClass improves plasmid sequence classification. *PLoS Comput. Biol.* **16**, e1007781 (2020).
5. *plasmid-segment-dispersal-2024*. (Github).
6. Shapiro, J. T. *et al.* Multilayer networks of plasmid genetic similarity reveal potential pathways of gene transmission. *ISME J.* (2023) doi:10.1038/s41396-023-01373-5.

REVIEWERS' COMMENTS

Reviewer #1 (Remarks to the Author):

I am satisfied with the revisions as such.

Reviewer #2 (Remarks to the Author):

Thank you for performing a very thorough revision of the manuscript and very informative additional comment in the accompanying letter. I recommend that the manuscript is accepted with no further revision.